# The Impact of a 12-Week Aqua Fitness Program on the Physical Fitness of Women over 60 Years of Age

**DOI:** 10.3390/sports12040105

**Published:** 2024-04-11

**Authors:** Katarzyna Kucia, Agnieszka Koteja, Łukasz Rydzik, Norollah Javdaneh, Arash Shams, Tadeusz Ambroży

**Affiliations:** 1Institute of Sports Sciences, University of Physical Education, 31-571 Kraków, Poland; katarzyna.kucia@awf.krakow.pl (K.K.); agnieszka.koteja@doctoral.awf.krakow.pl (A.K.); tadek@ambrozy.pl (T.A.); 2Department of Biomechanics and Sport Injuries, Kharazmi University, Tehran 14911-15719, Iran; njavdaneh68@gmail.com; 3Department of Physical Education and Sports Science, Boroujerd Branch, Islamic Azad University, Boroujerd 69151-36111, Iran; arash.sh.king@gmail.com

**Keywords:** aqua fitness, Senior Fitness Test, older women, wellness

## Abstract

Background: This study aimed to assess the impact of a 12-week Aqua Fitness program on the physical fitness of older women and emphasize sustainable health practices for aging populations. We focused on evaluating the program’s effectiveness, using the Senior Fitness Test to measure improvements in physical capabilities. Methods: An experimental research design was implemented, with 30 participants aged 60 and older. The participants were divided into a control group and an experimental group, each comprising 15 individuals. The control group received aqua Fitness exercises, and the experimental group received aqua fitness exercises and isometric (combined) exercises. Lower limb muscle strength, upper limb muscle strength, lower body flexibility, upper body flexibility, dynamic balance, agility, and endurance were assessed using the Senior Fitness Test. Assessments were conducted pre- and post-training. Results: For a comparison within the group, combined exercises (aqua fitness and isometric exercises) had a significant effect on lower limb muscle strength, upper limb muscle strength, lower body flexibility, upper body flexibility on the right side, dynamic balance, agility, and endurance. Aqua fitness exercises alone showed significant effects on upper limb muscle strength, lower body flexibility, and endurance and no significant effects on other variables. For the comparison between groups, no significant differences were found between the effects of aqua fitness exercises and combined exercises on lower limb muscle strength, upper limb muscle strength, lower body flexibility, upper body flexibility, and endurance. Significant differences were found only in dynamic balance and agility between the two groups of aqua fitness and combined exercises. Conclusions: Although the combined program (aqua fitness and isometric exercises) had a greater effect on improving the physical fitness of older adults than aqua fitness alone, there was no significant difference between the two groups. Therefore, the results of this study highlight the potential of aqua fitness in promoting sustainable health and physical fitness in the older adult population.

## 1. Introduction

Aging is a natural, irreversible process that affects 100% of the world’s population [1]. This process is accompanied by many phenomena that effectively hinder functioning in everyday life [2,3]. A loss of muscle mass, decreased cardiovascular fitness, and losses of balance and flexibility become real challenges for people in early old age [4,5]. Disorders like osteoarthritis, osteoporosis, decreased immunity, and decreased coordination or balance have been associated with aging, making the lives of patients more challenging compared to a healthy elderly population. Physically, locomotion and the activities of daily living become difficult, and vulnerability toward infections increases with aging [6,7,8].

This, in turn, can lead to a reduced quality of life, loss of independence, and increased risk of chronic diseases [9,10]. The aging of the population presents unique challenges and requires a systematic and sensible approach to maintaining physical fitness, thereby improving the overall health of older adults [11,12,13]. An important element of this prevention is the use of effective training methods that help older adults maintain or regain lost physical fitness [14,15,16,17]. Studies have shown that 46 to 51 percent of adults over age 60 are inactive, and about one-third are completely sedentary [18]. Physical activity for people over 60 is a recognized public health strategy [18] as evidence shows a reduced risk of all-cause mortality in people who adhere to the recommended levels of at least moderate–intensity physical activity [19,20,21,22]. Intervention design is one of the most influential factors in people’s participation in physical activity and wellness programs [23]. The World Health Organization (WHO)’s Member States and Partners for Sustainable Development Goals have created the Global Strategy and Action Plan for Ageing and Health for 2016–2020 and its continuation with the WHO program the Decade of Healthy Ageing 2020–2030. The WHO has established main priorities such as supporting country planning and action, collecting better global data, promoting research on healthy aging, aligning health systems to the needs of older people, laying foundations and ensuring that the human resources necessary for long-term integrated care are available, undertaking a global campaign to combat agism and enhancing the global network for age-friendly cities and communities [24].

Many exercise programs have been developed for older adults, and the heterogeneity of the different programs available is well recognized. Rodríguez et al. (2019) investigated the effects of three aerobic, strength, and aquatic exercise programs on older women. The results showed that women should consider performing aerobic activity up to age 60. After that age, aerobic activity or strength training can provide greater benefits [25]. Aqua fitness programs have also been the subject of numerous studies that have shown that this activity improves fatigue during the activities of daily living, reduces body fat percentage, improves physical functioning, coordination, and balance, and facilitates the socialization of older adults [26,27,28]. Based on a systematic review and meta-analysis in the field of aqua fitness programs, studies differ in terms of the statistical population, geography, measurement variables, comparison of protocols, intensity, and volume of exercise considered [28].

With the popularization of physical activity among older people, their awareness of a healthy lifestyle is increasing [29]. Regular exercise is now one of the key aspects of leading an active and satisfying life in early old age [30]. There is ongoing research to find the most optimal ways to be active in old age. One of the interesting areas that attracts the attention of scientists and trainers of older people is water training. An example of such a form of exercise is aqua fitness [31]. Aqua fitness training, which involves performing various and versatile activities in water (often with music), is one of the ways to activate seniors that is gaining increasing popularity [32]. Water provides an excellent training environment because it reduces joint stress, thereby minimizing the risk of overuse injuries. Additionally, aqua fitness combines aerobic, strength, and dance–gymnastic training, allowing for comprehensive physical fitness training [33]. However, little research has focused on the specific outcomes of this type of training in women over 60 years of age. Research on aqua fitness training and its impact on the functioning of older adults is important and topical, considering the increasing number of individuals worldwide. A study that evaluated the effect of two programs of walking on land versus walking in water [34] showed that a water exercise program can be considered a safer and preferable activity for older adults, providing body composition benefits similar to or higher than on-land exercises. Older participants also benefit from the lower impact forces and reduced fall risk associated with water walking without compromising their improved cardiorespiratory fitness [34]. Also, a meta-analysis examined the effects of water-based exercise programs on functional fitness in healthy older adults compared to other land-based physical exercise programs. The results showed that exercises in aquatic environments are effective in maintaining and improving functional fitness in healthy older adults. Furthermore, compared to exercise on land, exercise in an aquatic environment appears to be at least as effective and can be used as an alternative form of exercise [28]. Also, a meta-analysis has examined the effect of aquatic exercises on muscle strength in older adults. The results showed that land-based and aquatic exercises seem to lead to similar muscle strength gains. Aquatic exercise should be recommended as a strategy to improve muscle strength, but new studies with better methodological quality should be conducted [35].

Physical fitness is related to activity and, in old age, it is determined by a sufficient level of physical and motor independence [36]. Physical activity is of particular importance in the prevention of illness in older adults, maintaining independence, and improving quality of life. Despite the obvious benefits for older adults, for example, preventing collapse, maintaining independence, reducing isolation, and maintaining social relations to improve mental health, women and men are becoming less involved in physical activity with increasing age [37]. Therefore, this study aimed to evaluate the impact of a 12-week aqua fitness program on the physical fitness of older women.

## 2. Materials and Methods

### 2.1. Study Design

This study was experimental in nature and conducted at the Academy of Physical Education in Krakow, Poland. In accordance with the assumptions for conducting the pedagogical experiment, the researchers’ intervention involved manipulating the aqua fitness training process through the implementation of a standard training program supplemented with isometric exercises which consist of maximally tensing the muscles without changing their length (Table 1). To conduct the experiment, a technique involving working in two equally sized groups of the qualified group of women was applied. The project participants were randomly divided (randomization) into two groups, a control group and an experimental group, each consisting of 15 individuals.

The participants were partially blinded so that they were unaware of the hypothesized differences between the groups, but they were aware of which treatment they were assigned to. Computerized random numbers were generated to allocate participants. The allocation sequence was concealed from the researcher enrolling and assessing participants and arrived in sequentially numbered, opaque, sealed envelopes. Following enrolment, the participants were randomized (balanced) into either the control or experimental group. In the control group (Figure 1), aqua fitness training was performed according to the standard general cycle. In the experimental group, in addition to aqua fitness training, the participants performed isometric exercises. Changes were observed and evaluated throughout the experiment. The dependent variable in this case was physical fitness, understood as differences in the results of the Senior Fitness Tests performed in both groups. Evaluations of the participants were conducted before and after the training cycle. An important factor was also the ability to ensure high motivation for training during the experiment, which required high precision and the maximum engagement of the participants.

#### 2.1.1. Participants

This study was conducted on a group of 30 women aged 60 and older who were residents of the city of Krakow and its surroundings and qualified for the program based on the prevailing inclusion and exclusion criteria (Table 1) for diagnostic studies. Participants were recruited by sending messages via social networks and through advertisements that were displayed in entertainment, sports, and welfare centers. Individuals qualified for the project underwent medical examinations and were under medical supervision throughout the project. Participation in the project did not require sports experience, special motor competencies, or swimming skills. Participants were covered by accident insurance. Participation in the project was voluntary with the expression of appropriate written consent regarding participation in the classes and research. Comprehensive studies were conducted at the University School of Physical Education in Krakow on the impact of various types of physical activity, supervised by specialists, on many psycho-motor aspects in women aged over 60. This research was carried out as part of the Minister of Science and Higher Education program “Regional Initiative of Excellence” in 2019–2022, project number 022/RID/2018/19, with a budget of PLN 11,919,908. This paper presents a segment of a three-month study conducted as part of the aforementioned project No. 24/PB/RID/2021, titled: “Health benefits of Aqua Fitness training supported by vibrotherapy as a prevention of sarcopenia among older adults”, which received positive approval from the Bioethical Commission at the District Medical Chamber in Krakow (license No. 195/KBL/OIL/2022). This research was conducted in accordance with the Declaration of Helsinki.

#### 2.1.2. Inclusion and Exclusion Criteria

Inclusion and exclusion criteria are presented in Table 2.

### 2.2. Sample Size Estimate

The size and statistical power of the sample were calculated using G*Power software A 3.1.9.7 repeated-measures analysis of variance (ANOVA) with an interaction within–between factors was used; effect size—0.27; α-level—0.05; statistical power—0.80; number of groups—2; and number of measures—2 (pre- and post-intervention evaluations). Therefore, the initial size of the total sample was estimated at 30 participants.

### 2.3. Isometric Exercise Protocol

The experimental program included specialized exercises that were created for the aqua fitness training program. In accordance with the recommendations of the American College of Sports Medicine [38,39], all physical exercise programs were conducted by qualified physical exercise technicians (with a degree in sports science) who specialized in hydrogymnastics (instructor course-level). The isometric training in the water environment was based on maintaining muscle tension in a specific position (20 s) without changing muscle length. Similar to other studies, the exercise protocol involved performing isometric contractions with both legs for four sets of two minutes, with two minutes of rest between sets (14 min total) [40,41]. This is in contrast to dynamic training, which includes movements and muscle contractions. The progressive exercise program was developed based on sports medicine principles [42].

Below are the isometric exercises applied in the training program:Basic stance; one leg raised forward; “noodle (a foam float with a longitudinal shape)” held under the knee; bending and extending the leg at the hip joint.Basic stance; one leg raised forward; “noodle” held under the knee; abduction and adduction of the leg at the hip joint.Basic stance; right leg raised forward; “noodle” held under the knee; circling the leg at the hip joint.Straddle stance; arms bent forward; a gymnastic ball squeezed between the hands. Twists of the torso in the transverse plane.Straddle stance; arms bent forward; a gymnastic ball squeezed between the hands. Arm extensions and bends.Straddle stance; arms bent forward; a gymnastic ball squeezed between the hands. Circular movements of the arms down, forward, and bringing the arms back to the starting position.“Balance”; “betomic (a multifunctional tool for Aqua Fitness or rehabilitation)” squeezed between the knees. Hip twists to the right and left with alternating arm movements.“Balance”; “betomic” squeezed between the knees. Torso extension to lying on the back and return to the starting position.“Balance”; “betomic” squeezed between the knees. Torso extension to lying on the side and return to the starting position.

### 2.4. Procedure

The tool used to assess physical fitness was the Fullerton Senior Fitness Test (FSFT), which was performed twice, before and after applying the aqua fitness program. The FSFT, created by Robert E. Rikli and C. Jessie Jones at the Lifespan Wellness Clinic, California State University, Fullerton, consists of six movement tasks to diagnose functional fitness in four areas [43]. The test includes 6 trials which aim to determine physical parameters such as muscular endurance, mobility, dexterity, speed, body balance, motor coordination, reaction time, and flexibility.

➢The 30 s Chair Stand (30′CS) (standing up from a chair)—lower limb muscle strength; the patient repeats full stands from the sitting position. Repetitions are performed within 30 s with the arms crossed over the chest. The score is the number of completed chair stands in 30 s.➢The Arm Curl Test (ACT) (forearm flexion)—upper limb muscle strength; the patient flexes the forearm in 30 s. The test is conducted on the dominant arm side (or stronger side). Curl the arm up through a full range of motion, gradually turning the palm up (flexion with supination). The score is the total number of controlled arm curls performed in 30 s.➢Chair Sit and Reach (CS&R) (seated reach)—lower body flexibility; from a sitting position on a chair, the patient tries to reach the toes with the leg straight in the knee joint. The result in centimeters shows the distance between the fingers and the toes. The value can be negative when the patient is out of range of motion. The best score was recorded in centimeters.➢Back Scratch (BS) (back scratch in standing)—upper body flexibility; the patient tries to join the hands behind the back, leading one hand from the top, and the other from the bottom. The result given in centimeters indicates the distance between the middle fingers. The value may be negative when the patient reaches further than the fingertips. The best score was recorded in centimeters. The higher the score the better the result.➢Foot Up and Go (FU&G) (stand up and go)—dynamic balance and agility; the patient circles the cone in the shortest possible time at a distance of 2.44 m from the sitting starting position and returns to the starting position. The best time was recorded during two trials.➢The 2 min Step in Place (2-SinP) (marching in place) test—endurance; the participant stands up straight next to the wall while a mark is placed on the wall at the level corresponding to midway between the patella (knee cap) and iliac crest (top of the hip bone). The participant then walks in place for two minutes, lifting the knees to the height of the mark on the wall. Resting is allowed, and holding onto the wall or a stable chair is allowed. Stop after two minutes of stepping. The total number of times the right knee reached the level of the bar in two minutes was recorded.

FSFT 1 and FSFT 2 were conducted after a 10 min warm-up. Each person performed two trials for the movement tasks, with the best trial being assessed. The test allows for the assessment of the level of physical fitness. It provides a detailed analysis of the components, allowing for the proper planning of training for older adults and monitoring effects. The evaluation of uncomplicated movement tasks performed in the test is easy and quick to interpret and allows for a clear presentation of tasks and results to the patient, which plays an important role in rehabilitation as awareness of positive changes and cooperation motivate older adults to continue working on themselves. Other advantages of the test include the use of simple, everyday movement patterns that are safe for both less and more active older adults and the possibility of indirectly determining motor characteristics such as strength, flexibility, motor coordination, and endurance without the need for specialized equipment or a separate room.

### 2.5. Statistical Analysis

Statistical Package for the Social Sciences (SPSS) version 19.0 was used for the statistical analysis. The assumption of normality of distribution was verified using the Shapiro–Wilk test. A mixed-model repeated-measures ANOVA (RM-ANOVA) was used to determine between-subject variables. A paired t-test was used to evaluate intra-group differences. An independent *t*-test was used to compare the variables at baseline. The level of statistical significance was set at 0.05. Effect sizes (ESs) and 95% confidence intervals (CIs) were then calculated to provide a measure of clinical meaningfulness. The significance level was set at 0.05. Effect sizes for outcomes were also calculated as standardized mean difference (Cohen’s d) values from estimated marginal mean values and standard error estimates from the primary adjusted analysis and interpreted according to Cohen’s criteria (small ≤ 0.2; moderate = 0.5; large ≥ 0.8) [44].

## 3. Results

At baseline, there were no significant differences between groups in any of the demographic and physical fitness variables.

The results of the repeated-measures ANOVA showed that the effect of time for the 30 s Chair Stand, Arm Curl Test, Chair Sit and Reach, and 2 min Step in Place was significant. The effects of the group and time * group interactions for Foot Up and Go were significant, and no significant difference was observed for the 30 s Chair Stand, Arm Curl Test, Chair Sit and Reach, Back Scratch, and 2 min Step in Place tests.

For the comparison within the group, the combined exercises (aqua fitness and isometric exercises) had a significant effect on the 30 s Chair Stand test (lower limb muscle strength), but aqua fitness exercises alone did not have a significant effect on this variable. For the comparison between groups, no significant difference was observed between the aqua fitness exercises and combined exercises in the 30 s Chair Stand test (the interaction effect of group and time was not significant) (Table 3 and Table 4).

For the comparison within the group, aqua fitness exercises and combined exercises (aqua fitness and isometric exercises) had a significant effect on the Arm Curl Test (upper limb muscle strength). For the comparison between groups, no significant difference was observed between aqua fitness exercises and combined exercises on the Arm Curl Test (the interaction effect of group and time was not significant) (Table 3 and Table 4).

For the comparison within the group, aqua fitness exercises and combined exercises (aqua fitness and isometric exercises) had a significant effect on Chair Sit and Reach (lower body flexibility) on both the left and right sides. For the comparison between groups, no significant difference was observed between aqua fitness exercises and combined exercises on Chair Sit and Reach (the interaction effect of group and time was not significant) (Table 3 and Table 4).

For the comparison within the group, combined exercises (aqua fitness and isometric exercises) had a significant effect on the Back Scratch test (upper body flexibility) on the right side. However, aqua fitness exercises alone did not have a significant effect on this variable on the right side or the left side, and none of the exercises had a significant effect on this variable. In the comparison between groups, no significant difference was observed between the aqua fitness exercises and combined exercises on the Back Scratch test (upper body flexibility) (the interaction effect of group and time was not significant) on both the left and right sides (Table 3 and Table 4).

For the comparison within the group, the combined exercises (aqua fitness and isometric exercises) had a significant effect on the Foot Up and Go test (dynamic balance and agility). However, the aqua fitness exercises alone did not have a significant effect on this variable. For the comparison between the groups, a significant difference was observed between the aqua fitness exercises and combined exercises on the Foot Up and Go test (the interaction effect of group and time was significant) (Table 3 and Table 4).

For the comparison within the group, the aqua fitness exercises and combined exercises (aqua fitness and isometric exercises) had a significant effect on the 2 min Step in Place (endurance) test. For the comparison between groups, no significant difference was observed between the aqua fitness exercises and combined exercises on the 2 min Step in Place test (the interaction effect of group and time was not significant) (Table 3 and Table 4).

## 4. Discussion

The research results suggest that the 12-week aqua fitness program had a beneficial impact on the magnitude of physical fitness indicators assessed based on the Senior Fitness Test. The results of individual trials varied but indicated an overall improvement in physical fitness in women aged over 60 years after the training program in the water environment. Significantly improved results in upper limb strength and endurance were observed in both the control and experimental groups after completing the aqua fitness program. This indicates that sensitivity or susceptibility to movement occurs at any age [1]. It is noteworthy that the improvement was significantly greater in the experimental group. This may indicate that in general, aqua fitness training positively affects the strength and endurance of older women, but supplementing these sessions with isometric training increases the effectiveness of the exercises. Improvement in these aspects of physical fitness is important as it may contribute to improving quality of life and independence in functioning in later senior years [2,3,9,10]. The experimental group showed significantly higher results after completing the experimental water-based training program compared to the control group in terms of increased lower limb strength and upper body flexibility. This may suggest that the differentiating factor applied in the experimental group, which consisted of specially selected isometric exercises, has a beneficial impact on both the increase in strength and flexibility of older individuals [45]. Therefore, such exercises should be one of the elements of designing training programs in a water environment.

No significant statistical changes before and after the applied program were noted in either group in the area of lower body flexibility. This may be due to the fact that the initial level of flexibility in the participants was very high and difficult to improve and that aqua fitness focuses more on functional activity in water than on improving flexibility [9]. There is a need for further research on programs focused on improving flexibility in older individuals, particularly in a water environment [11].

In the experimental group, favorable statistically significant changes in dynamic balance and agility were observed in the results after the experiment and between groups. This may indicate that supplementing dynamic exercises in water with static strength exercises positively affects coordination due to better stability, confidence, and greater limb strength [12,13,14]. Balance and a higher level of agility can help achieve a higher level of functional fitness in older women [15].

The results of this study confirm the benefits of applying innovative physical activity programs in older adults. Aqua fitness training can thus positively affect the mobility, postural stability, and physical fitness of older individuals, helping them both maintain independence and reduce the risk of chronic diseases. Water provides an excellent training environment that reduces joint loads and minimizes the risk of injury, which is important for this age group. The literature emphasizes the benefits of physical activity combined with water resistance during movement in a water environment, such as increased muscle strength and endurance, improved flexibility and mobility, lower blood pressure, improved joint health, weight loss support, stress reduction, and a minimized risk of injury or fall [46,47].

Evidence suggests that physical exercise can be used to restore or maintain functional independence in older adults and may also potentially prevent, delay, or reverse frailty [48]. Physical exercise was shown to increase protein synthesis and muscle mass, as well as improve neural recruitment and muscle strength, as explained by neural and morphological adaptations [49]. Physical exercise can lower the risk of falling in older adults by averting muscle mass reduction and improving balance control. In particular, leg strength training seems to be crucial in preventing falls as lower limb weakness has been identified as a significant risk factor for falling [50]. Physical exercise is an effective treatment to improve balance and reduce fall rates in older adults [51].

In conclusion, this paper aimed to explore the widely known issue addressed in the sciences of physical culture and physical fitness regarding the effectiveness of aqua fitness training in maintaining health in older people. Although it is known that training yields biological effects, work is still ongoing to develop effective training programs dedicated to older individuals [29]. A trend in scientific research is the development of effective physical training programs focused on the prevention of lifestyle diseases and falls [30,31]. The applied 12-week aqua fitness program has a beneficial impact on some aspects of physical fitness in women aged over 60 years. These observations align with the existing experiences of scientists, confirming observations that even individuals over 60 years of age can improve their functional fitness [32,33].

The practical value of water-based activity programs for older adults is evident. The aquatic environment ensures safe and effective training conditions and minimizes injury risks and joint strain, thereby offering a sustainable approach to improving fitness for older adults. Therefore, one of the strengths of this study was safer training conditions for the participants. Another strength of this study was the examination of a large number of physical fitness factors including lower limb muscle strength, upper limb muscle strength, lower body flexibility, upper body flexibility, dynamic balance, agility, and endurance in older adults.

### Limitation

Although the Senior Fitness Test is designed for individuals who can move independently, there are certain limitations to the Fullerton Test. For their safety, individuals with severe musculoskeletal damage and balance disorders may not be able to fully perform the “2 min Step in Place” and “Stand Up and Go” tests. Additionally, for some people affected by coronary artery disease, an unstable blood pressure above 160/100 mmHg, or cardiac arrhythmias, the endurance test poses a high risk. It should also be emphasized that the results of this study are limited to the 12-week aqua fitness program, and further research is needed to assess the long-term effects of such a program. Moreover, the sample size is relatively small, which may affect the overall representativeness of the results. Also, the lack of male participants and the absence of a control group (inactive individuals) were limitations of this study. Another limitation of this study was the lack of control over the nutritional and medicinal conditions of the participants. Research on the long-term effects of aquatic fitness programs in the elderly population should continue. The sample size and duration of the program could affect the generalizability of the results. Therefore, another investigation involving a larger number of participants and a longer observation period is needed.

## 5. Conclusions

The following conclusions can be drawn based on the present research: The research results confirmed that aqua fitness classes are beneficial for women aged over 60 years. Participation in the aqua fitness program brought significant benefits in terms of improving physical fitness in this age group. Improvements were found in upper and lower limb strength, upper limb flexibility, agility, balance, and endurance. The application of the experimental program increased the effectiveness of the aqua fitness training in terms of increasing lower limb strength and upper limb flexibility and improving balance, agility, and endurance. The importance of physical activity among older individuals in various senior activation programs such as aqua fitness can contribute to maintaining independence, improving quality of life, and reducing the risk of chronic diseases.

## Figures and Tables

**Figure 1 sports-12-00105-f001:**
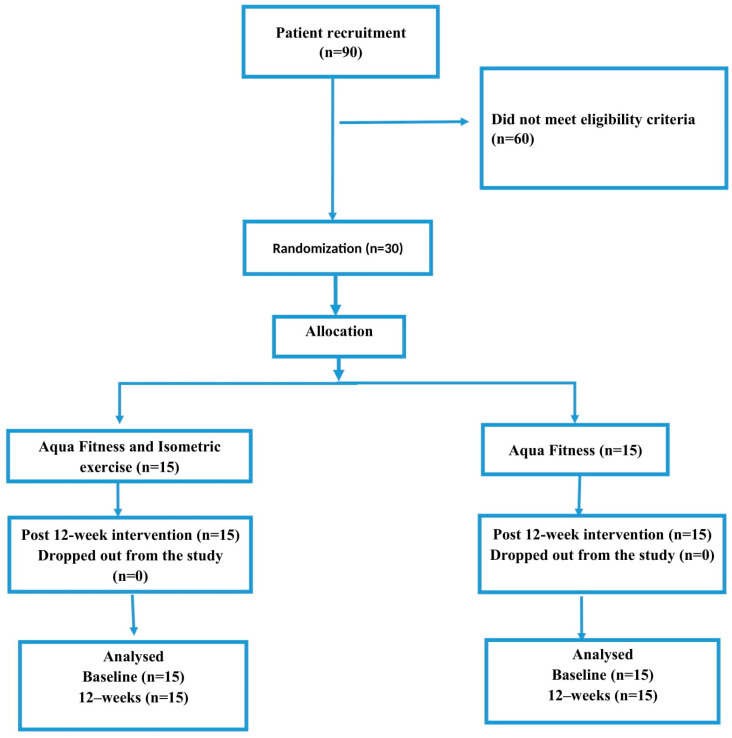
Flow diagram of participant recruitment.

**Table 1 sports-12-00105-t001:** Description of aqua fitness exercises.

Part of Classes	Contents	Duration
Introductory part	Greeting, checking attendance, and giving lesson assignments. Warming-up in the water. Choreography based on walking steps performed in different directions at a gradually increasing pace with gradually introduced arm work.	10 min
Main part	Strengthening exercises: Strengthening exercises were based on basic Aqua Aerobic steps, such as the following: - Balance (maintaining the body vertically in a squatting position using alternating work of the arms, moving in this position in different directions, and various movements of the legs and arms in all planes while maintaining the “balance” position); - Walking (alternating movements of the legs and arms in place and in different directions, modifications of arm work, changes in pace and positioning of feet and hands); - Running (alternating movements of legs and arms in place and in different directions, modifications of arm work, changes in pace and positioning of feet and hands); - Kicking (alternating kicks in different direction, with different paces and with different foot positions); - Jumping (jumping with both feet in place and with displacement); - Rocking (PW: standing with arms to the side and bent at the elbows. Movement: jumping from leg to leg while lifting the opposite knee up, making an alternating extension and bending the arms with different hand positions). In the experimental group, specialized isometric exercises designed for this purpose were added to the main part.	40 min
Final part	Calming, stretching, and relaxing exercises at the edge of the pool.	10 min

**Table 2 sports-12-00105-t002:** Inclusion and exclusion criteria for participating in the experiment.

Inclusion Criteria	Exclusion Criteria
Female gender	Previous surgeries in the last year
Age 60 years or older	Neurological deficits
Residence in or near Krakow	Cardiovascular diseases and pacemaker
No health contraindications to participate in aqua fitness classes, confirmed by a doctor	Cancers
Provided informed consent to undergo diagnostic tests and participate in the project	Trip to a sanatorium during the project
No injuries to the lower and upper limbs, or the last injury occurred more than 3 years before the start of the project	Scheduled surgery or hospitalization during the project
Regular participation in planned classes: 2 times a week for 12 weeks	Missing more than 20% of classes

**Table 3 sports-12-00105-t003:** Mean values of variable scores for between-group differences at baseline and within-group differences between pre- and post-program evaluations.

Variables	Group	Pre-Test Evaluation ^a^	Post-Test Evaluation ^a^	Within-Group Differences (Paired *t*-Test)		Between-Group Differences at Baseline (Independent *t*-Test)
				ES	(*p*-Value)	*p*
30CS [lp]	exp	15.26 ± 2.15	16.46 ± 1.76	0.61	(0.001) *	0.45
	con	13.73 ± 2.57	14.41 ± 3.86	0.29	(0.19)	
ACT [lp]	exp	17.33 ± 3.41	20.20 ± 3.44	0.84	(0.002) *	0.61
	con	16.26 ± 3.08	18.60 ± 2.69	0.81	(0.004) *	
CS&R Right [cm]	exp	8.23 ± 10.21	12.66 ± 8.51	0.47	(0.03) *	0.17
	con	−0.30 ± 16.62	6.43 ± 12.51	0.46	(0.016) *	
CS&R Right [cm]	exp	7.63 ± 10.43	13.90 ± 9.43	0.63	(0.010) *	0.078
	con	−1.66 ± 16.0	10.55 ± 11.23	0.88	(0.004) *	
BC Right [cm]	exp	0.66 ± 7.78	1.00 ± 7.55	0.75	(0.043) *	0.088
	con	5.28 ± 11.69	−3.03 ± 9.71	0.21	(0.094)	
BC Left [cm]	exp	−3.63 ± 5.50	−3.20 ± 6.43	0.88	(0.54)	0.10
	con	−10.30 ± 10.42	−8.66 ± 10.42	0.27	(0.15)	
FU&G [s]	exp	4.89 ± 0.66	4.59 ± 0.71	0.44	(0.002) *	0.58
	con	5.11 ± 0.72	5.44 ± 0.74	0.45	(0.10)	
2-sinP [lp]	exp	108.40 ± 10.22	118.86 ± 6.22	1.27	(0.008) *	0.43
	con	101.46 ± 16.40	113.80 ± 13.32	0.83	(0.001) *	

ES: effect size; ^a^: mean ± standard deviation; * statistically significant difference (*p* < 0.05). exp: experimental. con: control. ACT: Arm Curl Test; CS&R: Chair Sit and Reach; BC: Back Scratch; FU&G: Foot Up and Go; 2-sinP: 2-min Step in Place

**Table 4 sports-12-00105-t004:** Mean values of variable scores for between-group differences after the intervention.

Variables	Between-Group Differences (Repeated-Measures ANOVA)							
	Time			Time * Group		Group		
	*p*	ES	Mean Difference (%0.95 CI)	*p*	ES	*p*	ES	Mean Difference (%0.95 CI)
30′CS [lp]	0.009 *	0.22	−1.06 (−1.83 to −0.29)	0.72	0.004	0.079	0.10	1.66 (−0.20 to 3.54)
ACT [lp]	0.001 *	0.46	−2.60 (−3.67 to −1.52)	0.61	0.009	0.20	0.056	1.33 (−0.78 to 3.44)
CS&R Right [cm]	0.001 *	0.32	−5.58 (−8.72 to −2.44)	0.45	0.020	0.093	0.098	7.38 (−1.30 to 16.07)
CS&R Right [cm]	0.001 *	0.44	−7.56 (−10.81 to −4.31)	0.41	0.23	0.067	0.11	8.00 (−0.58 to 16.58)
BC Right [cm]	0.12	0.083	−1.29 (−2.94 to 0.36)	0.24	0.048	0.14	0.075	4.99 (−1.79 to 11.77)
BC Left [cm]	0.20	0.05	−1.03 (−2.68 to 0.61)	0.56	0.46	0.059	0.12	6.07 (−0.26 to 12.40)
FU&G [s]	0.86	0.001		0.005 *	0.25	0.032 *	0.15	−0.53 (−1.02 to −0.05)
2-sinP [lp]	0.001 *	0.61	−11.40 (−14.92 to −7.87)	0.59	0.010	0.15	0.071	6 (−2.37 to 14.37)

ES: effect size; CI: confidence interval; * statistically significant difference (*p* < 0.05). 30′CS: 30 s Chair Stand; ACT: Arm Curl Test; CS&R: Chair Sit and Reach; BC: Back Scratch; FU&G: Foot Up and Go; 2-sinP: 2-min Step in Place

## Data Availability

All data are included in the text of the manuscript.

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
