# Peer review of "The Impact of a 12-Week Aqua Fitness Program on the Physical Fitness of Women over 60 Years of Age"

_sports, 2024, doi:10.3390/sports12040105_

Round 1

Reviewer 1 Report (Previous Reviewer 1)

Comments and Suggestions for Authors

Abstract: “This study aimed to assess the impact of a 12-week Aqua Fitness program on the physical fitness of senior women”. It seems that your real aim was to compare the effects of a 12-week Aqua Fitness program alone and combined with isometric exercises in older women.

Abstract: “lower limb muscle strength”. As it is the beginning of a sentence, “Lower” should start with a capital letter.

Last paragraph of the introduction: In this paragraph, you stated the aim of the study. However, it is suggested that you rewrite it adapting to a more direct text.

For instance, instead of writing “The detailed aim of this article is to present the results of research on the impact of a 12-week Aqua Fitness program on the physical fitness of women over 60 and to evaluate the effects of this training using the Senior Fitness Test”, change to
This study aimed to evaluate the impact of a 12-week Aqua Fitness program on the physical fitness of older women.
Thus, leave the information regarding the methods, such as the use of the Senior Fitness Test, in the Materials and methods section. Furthermore, leave the specification of the participants’ age to the Participants and eligibility criteria section.

Moreover, I recommend that you erase the following part:
“The value of this research lies in demonstrating how Aqua Fitness can contribute to sustainable health and well-being in the context of aging, particularly for the female population. These actions are crucial for improving the quality of life of older adults, offering a proactive approach to counteract the negative effects associated with the loss of physical fitness in old age. Furthermore, this study aligns with the principles of Sustainable Development Goals by promoting healthy lifestyles and well-being at all ages (33). The results can provide valuable insights for both older individuals and professionals in health and recreation fields, aiding in the development of training programs that are tailored to individual needs and goals, and fostering a sustainable approach to health management in older populations.”

This part seemed quite biased. However, one of the premises of scientific research is to seek impartiality in data analysis. Therefore, leave to mention possible benefits of the exercise program in the Discussion section, with appropriate citation of references.

Materials and Methods: Separate the "Study design" section from information about the participants and their inclusion and exclusion criteria. So create a section called "Participants", for example. Likewise, create a separate section for ethical aspects.

“The dependent variable in this case was physical fitness, understood as differences in the conducted fitness trials in both groups using the Senior Fitness Test.” In this sentence, you present the concept of physical fitness, stating that physical fitness is understood as differences in the conducted fitness trials in both groups using the Senior Fitness Test. However, the concept of physical fitness should be in the Introduction of your study with the due reference.

In the Senior Fitness Test, use the already standardized acronyms referring to the tests. Moreover, check the name and the acronym used in the first sentence of the Procedure section. Fullerton Senior Fitness Test (FFST). Instead of FFST, change to FSFT.

Due to the characteristics of the study, it is suggested that authors check the CONSORT checklist. Thus, one of the improvements in the study is to include one figure with the flow chart of the sample selection process.

Isometric exercise protocol: You mentioned that “The isometric training in a water environment was based on maintaining muscle tension in a specific position (20 seconds)”. However, you should also specify the interval between each position. Or was it without interval? Moreover, add how much time (in minutes) this training session lasted. Also, why did you choose this training sequence? Is there any study to support this choice? I suggest that you include some reference(s) to support this training program.

Statistical Analysis: Add the interpretation and the reference regarding the effect size values.

Table 3: The presentation of the numbers in the table is divided, hindering reading. Consider placing the table horizontally rather than vertically to avoid this. Furthermore, remove the vertical lines from the table.

Comments on the Quality of English Language

Editing of English language and style required. Several grammatical issues need editing. Authors should seek additional assistance with translational services to improve the readability of this article.

Author Response

Dear Reviewer,

Thank you very much for your time and valuable comments, which all have been considered and incorporated. The detailed list of responses is given below. We hope that the modifications and explanation will be acceptable for you.

Yours sincerely,

Rydzik, corresponding author

Abstract: “This study aimed to assess the impact of a 12-week Aqua Fitness program on the physical fitness of senior women”. It seems that your real aim was to compare the effects of a 12-week Aqua Fitness program alone and combined with isometric exercises in older women.

Abstract: “lower limb muscle strength”. As it is the beginning of a sentence, “Lower” should start with a capital letter.

Corrected.

Last paragraph of the introduction: In this paragraph, you stated the aim of the study. However, it is suggested that you rewrite it adapting to a more direct text.
For instance, instead of writing “The detailed aim of this article is to present the results of research on the impact of a 12-week Aqua Fitness program on the physical fitness of women over 60 and to evaluate the effects of this training using the Senior Fitness Test”, change to
This study aimed to evaluate the impact of a 12-week Aqua Fitness program on the physical fitness of older women. Thus, leave the information regarding the methods, such as the use of the Senior Fitness Test, in the Materials and methods section. Furthermore, leave the specification of the participants’ age to the Participants and eligibility criteria section. Moreover, I recommend that you erase the following part:
“The value of this research lies in demonstrating how Aqua Fitness can contribute to sustainable health and well-being in the context of aging, particularly for the female population. These actions are crucial for improving the quality of life of older adults, offering a proactive approach to counteract the negative effects associated with the loss of physical fitness in old age. Furthermore, this study aligns with the principles of Sustainable Development Goals by promoting healthy lifestyles and well-being at all ages (33). The results can provide valuable insights for both older individuals and professionals in health and recreation fields, aiding in the development of training programs that are tailored to individual needs and goals, and fostering a sustainable approach to health management in older populations.”

This part seemed quite biased. However, one of the premises of scientific research is to seek impartiality in data analysis. Therefore, leave to mention possible benefits of the exercise program in the Discussion section, with appropriate citation of references.

Corrected.Therefore, This study aimed to evaluate the impact of a 12-week Aqua Fitness program on the physical fitness of older women. Additional content was removed.

Materials and Methods: Separate the "Study design" section from information about the participants and their inclusion and exclusion criteria. So create a section called "Participants", for example. Likewise, create a separate section for ethical aspects.

It was revised.

“The dependent variable in this case was physical fitness, understood as differences in the conducted fitness trials in both groups using the Senior Fitness Test.” In this sentence, you present the concept of physical fitness, stating that physical fitness is understood as differences in the conducted fitness trials in both groups using the Senior Fitness Test. However, the concept of physical fitness should be in the Introduction of your study with the due reference.

Added. Physical fitness is related to activity and its basis in old age is a sufficient level of physical and motor independence. Physical activity is of particular importance in the elderly in the prevention of illness, maintaining independence and improving the quality of life. Despite the obvious benefits for the elderly, for example, preventing collapse, remaining independence, reducing isolation and maintaining social relations to improve mental health, with age increasing, women and men are doing physical activity.

In the Senior Fitness Test, use the already standardized acronyms referring to the tests. Moreover, check the name and the acronym used in the first sentence of the Procedure section. Fullerton Senior Fitness Test (FFST). Instead of FFST, change to FSFT.
Corrected.
Due to the characteristics of the study, it is suggested that authors check the CONSORT checklist. Thus, one of the improvements in the study is to include one figure with the flow chart of the sample selection process.
Added.

Isometric exercise protocol: You mentioned that “The isometric training in a water environment was based on maintaining muscle tension in a specific position (20 seconds)”. However, you should also specify the interval between each position. Or was it without interval? Moreover, add how much time (in minutes) this training session lasted. Also, why did you choose this training sequence? Is there any study to support this choice? I suggest that you include some reference(s) to support this training program.

The exercises were 3 sets of 20 seconds of each exercise was performed with a 30 seconds break between sets and 1 minute before proceeding to the next exercise (a total of approx-imately 30 min per session). The progressive isometric exercise intervention was developed based on sports medicine principles (24).

Statistical Analysis: Add the interpretation and the reference regarding the effect size values.

 Effect sizes for outcomes were also calculated as standardized mean difference (Cohen's d) from estimated marginal mean and standard error estimates from the primary adjusted analysis, and interpreted according to Cohen's criteria (small ≤ 0.2; moderate = 0.5; large ≥ 0.8)

Table 3: The presentation of the numbers in the table is divided, hindering reading. Consider placing the table horizontally rather than vertically to avoid this. Furthermore, remove the vertical lines from the table.

The table was separated for better understanding.

Reviewer 2 Report (Previous Reviewer 4)

Comments and Suggestions for Authors

Dear all, I believe that the manuscript has been worked on extensively. The authors significantly improved the study, which in my opinion met the criteria to be published.

Author Response

Thank you

Reviewer 3 Report (Previous Reviewer 2)

Comments and Suggestions for Authors

I'm not sure why the article has been resubmitted with little change. The article corrections are still under tracked changes. The comments remain the same as my prior comments in the previous rejected version:

The title is very long 

No novel results, with plenty of similar papers in the literature.

Tiny sample size

The analysis is incorrect. You have compared pre and post within control and experimental arms, which is incorrect. There will be a difference when analysed this way due to the placebo effect. Firstly, baseline differences should be examined between control and experimental arms, which was not done. Then, the change is compared for the experimental versus the control group. Therefore, the interpretation of the results is incorrect. As such, I recommend rejection. 

Comments on the Quality of English Language

-

Author Response

Dear Reviewer,

Thank you very much for your time and valuable comments, which all have been considered and incorporated. The detailed list of responses is given below. We hope that the modifications and explanation will be acceptable for you.

Yours sincerely,

Rydzik, corresponding author

Comments and Suggestions for Authors
I'm not sure why the article has been resubmitted with little change. The article corrections are still under tracked changes. The comments remain the same as my prior comments in the previous rejected version:

The title is very long 
The title was shortened.

The Impact of a 12-Week Aqua Fitness Program on the Physical Fitness of Women Over 60 Years Old

No novel results, with plenty of similar papers in the literature.

 The introduction was modified and strengthened. The studied community, combined exercises and water exercises are among the outstanding features of this study.

Tiny sample size

 The size and statistical power of the sample were calculated using the G*Power software application. A repeated-measures analysis of variance (ANOVA) with interaction within-between factors was used; effect size: 0.27; α-level: 0.05; statistical power: 0.80; number of groups: 2; number of measures: 2 (pre- and post-intervention). Therefore, the initial size of the total sample was estimated at 30 participants.

The analysis is incorrect. You have compared pre and post within control and experimental arms, which is incorrect. There will be a difference when analysed this way due to the placebo effect. Firstly, baseline differences should be examined between control and experimental arms, which was not done. Then, the change is compared for the experimental versus the control group. Therefore, the interpretation of the results is incorrect. As such, I recommend rejection. 

Independent t-test was used to compare the variables in the baseline. At the baseline, there were no significant differences between groups in any of physical fitness variables.

The work has been revised by a native speaker 

Reviewer 4 Report (New Reviewer)

Comments and Suggestions for Authors

Dear Authors ,

The work needs to be corrected within the indicated sections:

Introduction:

Please add in the introduction a paragraph on the impact of aging on the increased risk of osteoporosis, decreased immunity or decreased coordination or balance. All of these changes can increase the risk of injury and thus increase the quality of life of the elderly.

Material and Methods:

- How were study participants selected , and what route were they invited to participate in the study?

- please include in this section a graph showing the course of the experiment including time intervals, group size, duration, etc.

- in the section describing the individual trials of the test it would be worthwhile to add some illustrations , photos

Results:

Please present part of the results in graphical form, which will be an interesting form for potential readers

- in tables please underline and highlight statistically significant results

Discussion:

Please elaborate on the paragraph confirming the fact that improving fitness, coordination or agility may reduce the risk of falls, which indirectly may have an impact on reducing the number of fractures, for example in the femur, which is very common in the elderly and is one of the factors of increased mortality and reduced quality of life of the elderly. Please support this with relevant literature

Author Response

Dear Reviewer,

Thank you very much for your time and valuable comments, which all have been considered and incorporated. The detailed list of responses is given below. We hope that the modifications and explanation will be acceptable for you.

Yours sincerely,

Rydzik, corresponding author

Comments and Suggestions for Authors
Dear Authors ,

The work needs to be corrected within the indicated sections:

Introduction:

Please add in the introduction a paragraph on the impact of aging on the increased risk of osteoporosis, decreased immunity or decreased coordination or balance. All of these changes can increase the risk of injury and thus increase the quality of life of the elderly.

Loss of muscle mass, decreased cardiovascular fitness, and loss of balance and flexibility become real challenges for people in early old age (4, 5). Disorders like osteoarthritis, os-teoporosis, decreased immunity, decreased coordination or balance have been associated with aging, making the life of patients more challenging as compared to a healthy elderly population. Physically, locomotion and daily activities become difficult, and vulnerability toward infections increases with aging (6-8).

Material and Methods:

- How were study participants selected , and what route were they invited to participate in the study?

Subjects were recruited through massage in social networks and through advertisements that were displayed in entertainment, sports and welfare centers.

- please include in this section a graph showing the course of the experiment including time intervals, group size, duration, etc.

Flow diagram of participants’ recruitment added.

Results:

Please present part of the results in graphical form, which will be an interesting form for potential readers

For better understanding, we tried to report the results in two tables.

- in tables please underline and highlight statistically significant results
Corrected.

Discussion:

Please elaborate on the paragraph confirming the fact that improving fitness, coordination or agility may reduce the risk of falls, which indirectly may have an impact on reducing the number of fractures, for example in the femur, which is very common in the elderly and is one of the factors of increased mortality and reduced quality of life of the elderly. Please support this with relevant literature.

Added. Evidence suggests that physical exercise can be used to restore or maintain functional independence in older adults, and may also potentially prevent, delay, or reverse frailty (De Vries et al., 2012). Physical exercise was shown to increase protein synthesis and muscle mass, as well as to improve neural recruitment and muscle strength, explained by neural and morphological adaptations (Guizelini, de Aguiar, Denadai, Caputo, & Greco, 2018). Physical exercise can lower the risk of falling in elderly people, averting muscle mass reduction, and improving balance control. In particular, leg strength training seems to be crucial in preventing falls, as lower-limb weakness has been identified as a significant risk factor for falling (Moreland, Richardson, Goldsmith, & Clase, 2004). Physical exercise is an effective treatment to improve balance and reduce fall rates in the elderly (Papalia et al., 2020).

Round 2

Reviewer 3 Report (Previous Reviewer 2)

Comments and Suggestions for Authors

I apologise to the authors for MDPI approaching me time and again in spite of my recommendation to reject. The comments remain the same as my prior comments in the previous rejected version, as the analysis is incorrect. You have compared pre and post within control and experimental arms, which is incorrect. There will be a difference when analysed this way due to the placebo effect. The change needs to be compared for the experimental versus the control group. Therefore, the interpretation of the results is incorrect. You have also clearly gone back with a n of 30 to fit numbers in the sample size calculation - I imagine that is why a random effect size of 0.27 was used (this is a very random number). The latest manuscript also reads poorly as the corrections have impacted the tables which are now unreadable.

As such, I recommend rejection. 

Comments on the Quality of English Language

-

This manuscript is a resubmission of an earlier submission. The following is a list of the peer review reports and author responses from that submission.

Round 1

Reviewer 1 Report

Comments and Suggestions for Authors

The article has some well-written parts. However, some long and complex sentences could be simplified to make reading easier.

The text of the Introduction presents information that should be included in the methods, such as ethical issues. Authors must use the Introduction to contextualize the topic and justify the gap that the study will fill.

In the Introduction, the authors state that "[...] study aligns with the principles of Sustainable Development Goals [...]". However, there is no reference supporting the Sustainable Development Goals.

The sample size is relatively small (30 participants), which may limit the generalizability of the results. Furthermore, I did not find information about sample size calculation in the text.

The authors state that "The project participants were randomly divided (randomization) into two groups [...]". It is highly recommended that the authors specify how the randomization process was conducted.

Due to the characteristics of the study, it is suggested that authors check the CONSORT checklist.

Comments on the Quality of English Language

Editing of English language and style required. 

Reviewer 2 Report

Comments and Suggestions for Authors

The title is very long

It is not very clear that the main difference between control and experimental arm was the use of isometric exercises. This should be made clear in the abstract and the intrpduction. 

The analysis is incorrect. You have compared pre and post within control and experimental arms, which is incorrect. There will be a difference when analysed this way due to the placebo effect. Firstly, baseline differences should be examined between control and experimental arms, which was not done. Then, the change is compared for the experimental versus the control group. Therefore, the interpretation of the results is incorrect. As such, I recommend rejection. 

Comments on the Quality of English Language

-

Reviewer 3 Report

Comments and Suggestions for Authors

Dear authors,

The objective of her manuscript is to demonstrate how aquafitness can benefit the physical function of older adult women evaluated through the Senior Fitness Test. The problem is that doing a quick search through the literature shows up many studies with the same objective. For example:
- Leirós-Rodríguez, R., Romo-Pérez, V., Pérez-Ribao, I., & García-Soidán, J. L. (2019). A comparison of three physical activity programs for health and fitness tested with older women: Benefits of aerobic activity, aqua fitness, and strength training. Journal of Women & Aging, 31(5), 419-431.
- Leirós-Rodríguez, R., Soto-Rodríguez, A., Pérez-Ribao, I., & García-Soidán, J. L. (2018). Comparisons of the health benefits of strength training, aqua-fitness, and aerobic exercise for the elderly. Rehabilitation research and practice, 2018.

And even a systematic review where different articles appear with the same topic and even with the same evaluation method:
- Waller, B., Ogonowska-Słodownik, A., Vitor, M., Rodionova, K., Lambeck, J., Heinonen, A., & Daly, D. (2016). The effect of aquatic exercise on physical functioning in the older adults: a systematic review with meta-analysis. Age and aging, 45(5), 593-601.

So I would like to know what your study provides new that is not described in the literature.

Furthermore, another aspect that is not described in the results of their study is the difference or not that existed between groups during the baseline.

Best regards.

Reviewer 4 Report

Comments and Suggestions for Authors

Dear all, I thank you for the opportunity to read and review this study. Here are my considerations:

Introduction

1. I suggest that information about ethical procedures and the location of the study be included in the Methodology section;

2. The theoretical framework still lacks more information about the benefits of Aqua Fitness programs for older adults. Especially, with the inclusion of quantitative data on changes/benefits to motor/physical abilities;

3. The section still lacks a good justification for carrying out the study. This means that authors must present, based on previous international studies or in their country, a convincing reason for carrying out the study, and consequently, publishing the results.

Methodology

1. I suggest anticipating the "Study Design" subsection and then presenting the "Participants" subsection;

2. Aqua Fitness training program must be unlinked from the "Study design" section;

3. The presentation of the type of study must prevail at the beginning of the "Study Design" section;

4. The table with information on inclusion and exclusion criteria is a strong point of the study;

5. Instruments:

Question: the authors did not evaluate BMI, comorbidities (self-report), number of medications? This is basic information so that readers know the health conditions of the participants... If this has not been done, I suggest presenting it as a "limitation of the study";

6. Statistical procedures:

Question: The authors calculated baseline differences using the Student T test, correct! And then, were the Post-test differences also verified by the same test? If so, there is a problem! This comparison should be carried out using another test.

Results

1. I suggest that all results are presented in a single table and not cut throughout the text. This makes it difficult to visualize general outcomes. I suggest searching previous articles, including Sports magazine itself, for similar previous studies.

Discussion

1. Finally, in the limitations, I also suggest including strengths of the study and suggestions for future research. If the authors do not make positive points in their study and are also unable to suggest future experiments, this is a serious limitation of the research group!

Conclusion

1. I suggest presenting the text in a single paragraph and not in points (1, 2, 3...).

Comments on the Quality of English Language

Dear Editor, I thank you for the opportunity to review the study. I consider that the study still needs corrections and adjustments. Thus, after the necessary corrections, it will still be possible to verify the improvement of the text. Among the merits of the research I consider the country of origin and the type of intervention (Aqua fitness). In general, there is a lack of information about this work methodology. On the other hand, the authors have not yet managed to express this in their paper.

Sincerely, MdMN